# Electrospinning Mo-Doped Carbon Nanofibers as an Anode to Simultaneously Boost Bioelectrocatalysis and Extracellular Electron Transfer in Microbial Fuel Cells

**DOI:** 10.3390/ma16062479

**Published:** 2023-03-21

**Authors:** Xiaoshuai Wu, Xiaofen Li, Zhuanzhuan Shi, Xiaohai Wang, Zhikai Wang, Chang Ming Li

**Affiliations:** Institute of Materials Science and Devices, School of Materials Science and Engineering, Suzhou University of Science and Technology, Suzhou 215011, China

**Keywords:** electrospinning, microbial fuel cells, Mo-doped carbon nanofibers, interface modification

## Abstract

The sluggish electron transfer at the interface of microorganisms and an electrode is a bottleneck of increasing the output power density of microbial fuel cells (MFCs). Mo-doped carbon nanofibers (Mo-CNFs) prepared with electrostatic spinning and high-temperature carbonization are used as an anode in MFCs here. Results clearly indicate that Mo_2_C nanoparticles uniformly anchored on carbon nanowire, and Mo-doped anodes could accelerate the electron transfer rate. The Mo-CNF ΙΙ anode delivered a maximal power density of 1287.38 mW m^−2^, which was twice that of the unmodified CNFs anode. This fantastic improvement mechanism is attributed to the fact that Mo doped on a unique nanofiber surface could enhance microbial colonization, electrocatalytic activity, and large reaction surface areas, which not only enable direct electron transfer, but also promote flavin-like mediated indirect electron transfer. This work provides new insights into the application of electrospinning technology in MFCs and the preparation of anode materials on a large scale.

## 1. Introduction

Energy shortage, environmental pollution, and water scarcity are urgent challenges for humans, and microbial fuel cells (MFCs) are increasing attention as an ecofriendly and sustainable energy conversion technology [1,2]. MFCs use electroactive bacteria as catalysts to convert chemical energy from organic substrates into electrical energy, and they are widely used in wastewater treatment, heavy-metal remediation, and other applications [3,4]. However, the sluggish extracellular electron transfer (EET) rate of MFCs leads to their low actual output power density, which limits large-scale applications [5]. The surface properties and structure of anodes are critical for increasing the available effective surface for bacterial colonization and interfacial electrocatalytic activity, and thus for improving electron transfer to solid electrodes [6,7].

Carbon material is still the most suitable anode material, with almost all the basic requirements such as high electrical conductivity, biocompatibility, and low cost [8,9]. Electrospinning is a direct and simple method for producing continuous nanofibers on a large scale [10]. Obtained fibers with electrospinning have the advantages of a small diameter and a large specific surface area [10]. After stabilization and carbonization, carbon nanofibers are obtained, which are widely used in many fields [11,12,13]. Therefore, many researchers are studying the use of electrospinning technology to enhance the performance of MFCs, and the outcomes show that electrospinning has a broad application prospect in the preparation of efficient microbial fuel cell electrodes [8,9,14]. For example, a self-supporting electrode composed of iron cobalt bimetallic metal–organic frameworks (FeCo-MOF/CNFs) prepared with electrospinning dramatically enhanced the output power of MFCs [15]. Nitrogen-doped carbon nanofibers anchored with iron nanoparticles (Fe/N-x@CNFs) were designed as anode electrocatalysts with good electrocatalytic activity and biocompatibility using electrospinning and calcination processes [16]. Molybdenum carbides with excellent electrocatalytic activity and good biocompatibility were extensively investigated for MFC anode modification [17,18]. For example, Zou et al. prepared molybdenum-carbide-hybridized graphene nanocomposite (Mo_2_C@G) with electrostatic self-assembly and high-temperature carbonization [17], synergistically promoting microbial biofilm growth and interfacial bioelectrocatalysis via Mo_2_C@G anodes for bioelectricity production. Molybdenum doping can improve the active site, electrocatalytic activity, and biocompatibility of carbon nanofibers. The preparation of Mo-doped carbon nanowires using electrostatic spinning improves the electrocatalytic activity and biocompatibility of the electrodes, and the nanowire structure adds more bacterial attachment sites. However, to the best of our knowledge, no one has prepared Mo-doped carbon nanowires using the electrostatic spinning technique for application in the anode of microbial fuel cells.

We prepared Mo_2_C-modified carbon nanowires as an anode for *Shewanella putrefaciens* CN32 (*S. putrefaciens* CN32) MFCs with a simple method. The material was specifically prepared via electrostatic spinning, followed by stabilization and carbonization. Electrochemical analysis confirmed that Mo-CNF ΙΙ exhibited high electrocatalytic activity towards flavin-like electron mediators. The doping of Mo_2_C improves the bacteria-to-electrode interfacial electron transfer, which enhances the biocompatibility of the electrode surface, adds more active sites, and increases electrocatalytic activity. The effects of different amounts of Mo doping on enhanced EET during bioelectrocatalysis were compared. MFCs equipped with Mo-CNF ΙΙ exhibited a great output power density of around 1287.38 mW·m^−2^.

## 2. Materials and Methods

### 2.1. Preparation of Mo-Doped Carbon Nanofibers

The preparation process of Mo-doped carbon nanofibers is shown in Figure 1. First, 1g of polyacrylonitrile (PAN) powder was weighed and added to N,N-dimethylformamide (DMF) to prepare a 10 wt % PAN solution, and a uniform solution was obtained via magnetic stirring for 24 h at room temperature. Next, molybdenum acetylacetone (MoO_2_(acac)_2_) was added and stirred for 24 h to obtain a green, homogeneous spinning precursor solution. Nanofiber membranes were fabricated with electrospinning. Electrospinning parameters were as follows: positive voltage, 8–10 KV; negative voltage, −3 to −2 KV; pushing speed, ~0.0008–0.001 mm/s; distance between needle and collection drum, 16–18 cm; rotational speed of the drum receiver, 100 rpm; temperature and humidity, controlled at 25 ± 3 °C and 35 ± 5%, respectively. First, the fiber membrane was preoxidized in a muffle furnace at 280 °C for 2 h with a heating rate of 5 °C/min to stabilize the fiber structure. It was then carbonized at 900 °C for 2 h in an argon atmosphere with a heating rate of 2 °C/min. After carbonization, the morphology of the fiber membrane was basically unchanged, and Mo-doped carbon nanofibers were obtained after grinding. When the MoO_2_(acac)_2_ addition levels in the spinning precursor solution were 0, 5, 10, and 20 mM, the prepared Mo-doped carbon fibers were designated as CNFs and Mo-CNFs I–III, respectively. Transmission electron microscopy (TEM, JEOL JEM-2100HR, Tokyo, Japan) and scanning electron microscopy (SEM, JEOL JSM-7800F, Tokyo, Japan) were used to analyze the morphology and structure of the material. The surface properties of the material were characterized with X-ray photoelectron spectroscopy (XPS) and Raman spectroscopy. The crystal structure and crystal phase of the material was determined with powder X-ray diffraction (XRD).

### 2.2. Electrochemical Characterization

An electrochemical workstation (CHI660, Shanghai Chen Hua Instrument Co., Ltd., Shanghai, China) was used for all electrochemical testing. A three-electrode system was used in the experiment with a carbon cloth modified with nanomaterials as the working electrode, saturated calomel electrode (SCE) as the reference electrode, and a titanium sheet (2 × 2 cm) as the counter electrode. First, the experiment was conducted in a 0.1 M phosphate buffer saline (PBS) buffer with 2 μM flavin mononucleotide (FMN). With the addition of FMN, cyclic voltammetry (CV), differential pulse voltammetry (DPV), and electrochemical impedance spectroscopy (EIS) experiments were conducted at regular intervals. When the peak current reached its maximum, CV tests with various sweep speeds were conducted. Subsequently, tests were performed in bacterial suspensions (*S. putrefaciens* CN32 suspended in M9 buffer (KH_2_PO_4_, NH_4_Cl, NaCl, Na_2_HPO_4_·12H_2_O) with lactic acid as the electron donor) by discharging at 0.2 V for 12 h and then testing CV, DPV, and EIS. The potential ranges for CV and DPV tests were −0.8 to 0.6 and −0.8 to 0 V, and the CV sweep rates were 1 and 30 mV s^−1^, respectively. The DPV test’s frequency was 1 Hz, the amplitude was 50 mV, and the potential step was 4 mV. All EIS experiments used a perturbation signal of 5 mV at −0.45 V and a frequency range of 0.01–100,000 Hz.

### 2.3. MFC Setup and Operation

In this work, *S. putrefaciens* CN32 was used as an electroactive microorganism, and the MFC’s performance was tested using an H-type double-chamber apparatus. H-type MFCs consist of two counter-mouth glass vessels separated by a proton exchange membrane (Nafion 117, N117), with the anode being a carbon cloth coated with nanomaterials and the cathode being a carbon brush. The cathodic solution was a 0.01 mM PBS buffer containing 50 mM potassium ferricyanide (K_3_Fe[(CN)_6_]), and the anodic solution was an M9 buffer with suspended *S. putrefaciens* CN32 and lactic acid as the sole electron donor. The discharge curves of MFCs were tested with an external 2 kΩ resistor for stable operation at 30 °C. After three rounds of discharge testing, and when the output voltage was stable, the resistance was changed (from 80,000 to 500 Ω) via the external resistor box to test the power-density and polarization curves.

After three rounds of discharge testing, the anodic biofilm treatment was first fixed with paraformaldehyde followed by gradient dehydration with ethanol. The concentrations of ethanol were 30%, 40%, 50%, 60%, 70%, 80%, 90%, and 100%, and each concentration was dehydrated for 20 min. In this paper, several steps were taken to prepare the electrode: 2 mg of material was put into a mortar, and 30–40 μL of a diluted polytetrafluoroethylene (PTFE) emulsion was added and mixed thoroughly to form a paste that was evenly coated on the front and back sides of the carbon cloth (1 × 1 cm), and dried under vacuum at 110 °C for 3 h to obtain the required electrode for the experiment. The details of the above operations and bacterial culture are in our previous work [19,20].

## 3. Results and Discussion

The detailed structural and morphological characterization of the Mo-CNFs and CNFs was examined with SEM and TEM. Figure 1a,b show that the CNFs were nanowires with a uniform diameter and a smooth surface, and Mo was uniformly distributed on the nanofiber surface in the form of nanoparticles in Mo-CNF ΙΙ. There was no significant change in the fiber diameter before and after Mo doping, and the fiber diameter was around 400–500 nm. Figure 1c,d show the TEM image and EDX element mappings of Mo-CNF ΙΙ. The results are consistent with SEM, and distinctly indicate that Mo elements were uniformly and effectively distributed within the entire nanofiber. Mo doping on the nanofiber surface increased fiber surface roughness and the electrochemically active sites, promoting more bacterial adhesion to the anode. This greatly improved the efficiency of extracellular electron transfer between the electrodes and bacteria.

The surface properties and crystal structure of the materials were characterized using XPS, Raman spectroscopy, and XRD. In the XRD spectrum (Figure 2a), two broad peaks at 25° and 43° were found for CNFs that were identified as graphitized carbon peaks [21]. The positions and relative intensities of the characteristic diffraction peaks in the Mo-CNFs were in good agreement with the PDF cards (PDF #15-0457), indicating that Mo was present in the carbon nanowires in the form of Mo_2_C. In addition, the characteristic peaks in Mo-CNF Ι were not obvious, probably because the Mo content was too small. The Raman analysis of Mo-CNF and CNF electrodes is shown in Figure 2b. The obvious peaks that existed at 1350 and 1580 cm^−1^ for all electrodes corresponded to the D and G peaks of the carbon material, respectively. The I_D_:I_G_ value is a crucial parameter for determining the degree of lattice collapse and graphitization. Analyzing the statistics showed that the I_D_:I_G_ values of Mo-CNFs were all higher than those of CNFs (0.94). Furthermore, Mo-CNF ΙΙ had the largest I_D_:I_G_ value of 0.97. This indicates that there were more lattice defects in Mo-CNF ΙΙ, which allowed for exposing more active centers and increasing electrocatalytic activity. The full XPS spectra in Figure 2c show that the surface of the Mo-CNFs contained four elements, namely, Mo (BE = 232.02 eV), N (BE = 398.03 eV), O (BE = 531.01 eV), and C (BE = 287.01 eV), compared to the surface of CNFs, demonstrating the effective doping of Mo into the carbon nanofibers [22]. Then, the 3d Mo spectrum (Figure 2d) showed that, in Mo-CNF ΙΙ, Mo3d could be divided into four peaks corresponding to MoO_x_3d_3/2_ (235 eV), Mo^2+^3d_3/2_ (232.63 eV), MoO_x_3d_5/2_ (231.67 eV), and Mo^2+^3d_3/2_ (228.82 eV) [23,24]. The high-resolution C1s, O1s, and N1s XPS spectra of Mo-CNF II are shown in Appendix A; the C1s spectrum shows that C1s could be divided into four peaks in Mo-CNF ΙΙ corresponding to C-N (285.47 eV), C-O (286.62 eV), C=O (289.15 eV), and C-C (284.57 eV) [25].

To comprehend the interfacial redox response of flavin media on the surface of various Mo-CNFs, we used a three-electrode system to perform the experiments in a PBS buffer containing 2 μM FMN. The CVs results show that all electrodes had oxidation peaks at the −0.45 V potential, which was the electrochemical response to the oxidation of flavin mediators on the electrode interface, as shown in Figure 3a. In addition, the peak current of CV curves increased and then decreased with the increase in Mo content, showing a volcanic curve. Mo-CNF ΙΙ exhibited the highest peak current density of 306.1 μAcm^−2^, which was much larger than that of the unmodified CNF electrode (84.2 μAcm^−2^). The results of the DPV curves (Figure 3b) are consistent with the CV curves. The maximal redox peak current of the CNF ΙΙ electrode in the DPV curve was 0.91 mA cm^−2^, which means that the surface of the Mo-CNF ΙΙ electrodes had the largest number of electrochemically active sites, which is favorable for promoting flavin-mediated indirect electron transfer. Figure 3c shows the EIS results that indicate that the *R*_ct_ of Mo-CNF I was much lower than that of the CNFs (123.36 Ω), and Mo-CNF ΙΙ had the lowest interfacial electron transfer impedance (5.59 Ω). In addition, the *R*_ct_ of Mo-CNFs Ι and ΙΙΙ were 12.08 and 9.55 Ω, respectively. This demonstrates that the modification of electrode interfaces with Mo_2_C considerably impacted the improvement in electron transfer rate. The relationship between peak current and sweep speed for CV curves with different sweep speed levels is shown in Figure 3d and Appendix A. The redox peak current of the CNF electrode tended to be more linearly related to the sweep rate, indicating that the electrochemical process was surface-controlled. In contrast, the redox peak current of the Mo-CNF ΙΙ electrode surface tended to be more linearly related to the square root of the sweep speed, showing that the electrochemical process was diffusion-controlled. The results show that the Mo-doped carbon nanowires transformed the interfacial electrochemical reaction from a surface-controlled process into a diffusion-controlled process, and promoted the rapid electrochemical reaction on the electrode surface.

Subsequently, the electrochemical properties of the Mo-CNFs and CNFs were evaluated in the *S. putrefaciens* CN32 cell suspension with 18 mM lactate, which served as the electron donor. First, electrodes were allowed to discharge at 0.2 V for 12 h, and the outcomes (Figure 4b) indicated that the discharge curves of the Mo-CNF anodes rose first and then remained stable. The Mo-CNF ΙΙ anode had the maximal discharge current (422.60 μA cm^−2^), while the CNF discharge current was 165.80 μA cm^−2^. Then, a CV test was conducted with a sweep speed of 1 mV s^−1^, and the outcomes are depicted in Figure 4a. The results show that all CVs had steady-state catalytic currents at −0.45 V. The −0.45 V position corresponded to the catalytic peak of the indirect electron transfer (IET) mediated by bacterial autocrine flavin-like electron mediators. Furthermore, Mo-CNF ΙΙ had the highest catalytic current (0.75 mAcm^−2^), while that of CNFs was 0.22 mAcm^−2^. The DPV curves displayed redox peaks at the 0 and -0.45 V potentials, as displayed in Figure 4c. The 0 V position corresponded to the catalytic peak of direct electron transfer (DET). The Mo-CNF ΙΙ electrode had the maximal redox peak current density at both the 0 and −0.45 V potentials, which agreed with the results of the CVs. Figure 4d depicts the EIS results, which show that the *R*_ct_ of Mo-CNFs was much lower than that of CNFs (598.20 Ω), and that Mo-CNF ΙΙ had the lowest interfacial electron transfer impedance (13.34 Ω). The results show that the Mo_2_C modification of the electrode interface could remarkably accelerate the electron transfer rate between bacteria and the electrode.

To assess the bioelectrocatalytic performance of the Mo-CNFs and CNFs, they were employed as anodes in H-type double-chamber MFCs inoculated with *S. putrefaciens* CN32 cell suspension. Figure 5a shows that the Mo-CNFs and CNFs MFCs were discharged with an external 2 kΩ resistor. The MFCs’ output current density using the Mo-CNF ΙΙ anode was stable at 0.20 mA cm^−2^ in the third discharge cycle, while the CNFs’ anode was 0.14 mA cm^−2^, demonstrating that the Mo_2_C-modified carbon nanofibers remarkably improved the bioelectrical output. After three rounds of discharge, the polarization and power-density curves of the anode were calculated by changing the resistance value of the external resistor (80,000–500 Ω) when the voltage was stable. Figure 5b shows that the maximal power densities of Mo-CNFs Ι–ΙΙΙ were 1019.70, 1287.38, and 986.05 mW m^−2^, respectively, which were significantly greater than the output power density of the undoped CNFs (649.69 mW m^−2^). Moreover, Mo-CNF ΙΙ had a maximal power density that was twice that of the CNFs. We compare the performance of the anode materials prepared in this work with previously reported anode materials in Appendix A. Figure 5c,d show the results of the characterization of Mo-CNF and CNF anode biofilms after three rounds of discharge. The results show that the Mo-CNF ΙΙ anode was enriched with a large number of bacteria, forming a good biofilm, but the microorganisms on the surface of the CNF anode were scarce. The improved biocompatibility of molybdenum-doped carbon nanofibers can be attributed to the good biocompatibility of Mo_2_C. Moreover, the presence of the Mo_2_C nanoparticles increased the number of active sites and promoted the adsorption of extracellular bacterial proteins. The aforementioned investigations demonstrate that the Mo-doped nanowires enhanced the anode’s biocompatibility and raised its level of bioelectric output.

## 4. Conclusions

In this study, an Mo_2_C nanoparticle-modified nanowire was obtained via electrospinning and high-temperature carbonization. Then, it was employed as an MFC anode to boost the generation of bioelectricity. The Mo-CNF IΙ anode in the MFCs achieved the maximal power density of 1287.38 mW m^−2^, which was double that of the unmodified CNF anode. The doping of Mo_2_C improved the bacteria-to-electrode interfacial electron transfer, which enhanced the biocompatibility of the electrode surface, added more active sites, and accelerated the direct electron transfer process. Additionally, Mo-CNF ΙΙ had a good response to FMN and promoted a flavin-like mediated indirect electron transfer. This work provides new insights into the application of electrostatic spinning technology in microbial fuel cells and the preparation of anode materials on a large scale.

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
