# Peer review of "Electrospinning Mo-Doped Carbon Nanofibers as an Anode to Simultaneously Boost Bioelectrocatalysis and Extracellular Electron Transfer in Microbial Fuel Cells"

_materials, 2023, doi:10.3390/ma16062479_

Round 1

Reviewer 1 Report

The manuscript deals with electrospinning Mo-doped carbon nanofibers to be used as an anode in microbial fuel cells. However, Molybdenum has been reported before, but using electrospinning and the carbonization technique makes the manuscript of matter of interest. So, it is highly recommended for publication after clarifying the following points.

1- The introduction section needs to be improved,  authors mentioned the benefits of using electrospinning technology to enhance the performance of MFCs, but didn't mention the importance of adding Mo- to CNFs

2- A comparison table or graph to compare the performance of this anode material with commercial and previously reported works is needed.

3- In the preparation, Why PAN was chosen, will any other polymer work??

4- Do the authors studied the effect of changing the electrospinning parameters on the structure or morphology of the produced fibers and therefore their effect on the performance ?.

5- Did the authors measure the SSA of the produced material and what about the long stability test??

Author Response

Dear Reviewers,

I would like to express my deep thanks to you for your constructive suggestions and comments for improving the manuscript. According to your comments, we carefully revised the manuscript and highlighted the changes in blue color. The detail responses to your comments are as follows. If further revisions are needed, please let us know.

With best regards,

Dr Xiaoshuai Wu

Suzhou University of Science and Technology, China

Email: wuxiaoshuai365@163.com

The manuscript deals with electrospinning Mo-doped carbon nanofibers to be used as an anode in microbial fuel cells. However, Molybdenum has been reported before, but using electrospinning and the carbonization technique makes the manuscript of matter of interest. So, it is highly recommended for publication after clarifying the following points.

Point 1: The introduction section needs to be improved, authors mentioned the benefits of using electrospinning technology to enhance the performance of MFCs, but didn't mention the importance of adding Mo- to CNFs

Response: Many thanks to reviewers for their valuable suggestions and comments. We further explain the importance of adding Mo to CNFs in row 50-55.

Point 2: A comparison table or graph to compare the performance of this anode material with commercial and previously reported works is needed.

Response: Many thanks for the reviewer’s comments and suggestions. We add the Table S1(Supporting Information) to compare the performance of the anode materials prepared by this work with commercial and previously reported anode materials, such as follows:

Anode materials

Reactor

configuration

Substrate

Source of

inoculation

Performance

Pmax (mW m-2)

Reference

Mo-CNFs ΙΙ

Dual chamber

M9 + Lactate

S.putrefaciens CN32

1287.38 mW m-2

This work

Fe@CNFs-20

Dual chamber

M9 + Lactate

S.putrefaciens CN32

731.05 mW m-2

[1]

CNTs/CNFs

Dual chamber

artificial wastewater + PBS

mixed

bacteria

362 ± 20 mW m-2

[2]

GL-MoS2/CC

Dual chamber

PBS + sodium acetate

mixed bacteria

960.4mW m−2

[3]

Mo2C@CF

Dual chamber

Lactate

S.putrefaciens CN32

1052mW m−2

[4]

Mo2C/CCT

Single chamber

sodium acetate

mixed bacteria

1120 mW m−2

[5]

MnCo2O4@CF

Dual chamber

sodium lactate

mixed bacteria

945 mW m−2

[6]

PANI-T-900

Dual chamber

Wastewater + acetate

mixed bacteria

40.4 mW m−2

[7]

CNPs/Ti

Single chamber

sodium acetate

mixed bacteria

616 mW m−2

[8]

Mo2C/CNTs

Single chamber

PBBM + glucose

E. coli

1050±26.4 mW m−2

[9]

pristine CC

Dual chamber

M9+sodium lactate

Shewanella oneidensis MR-1

51.5 mW m−2

[10]

  1. Sun, X.; Wu, X. S.; Shi, Z. Z.; Li, X.; Qian, S.; Ma, Y.; Sun, W.; Guo, C.; Li, C. M., Electrospinning iron-doped carbon fiber to simultaneously boost both mediating and direct biocatalysis for high-performance microbial fuel cell. J. Power Sources 2022, 530, 231277.
  2. Cai, T.; Huang, M. H.; Huang, Y. X.; Zheng, W., Enhanced performance of microbial fuel cells by electrospinning carbon nanofibers hybrid carbon nanotubes composite anode. Int. J. Hydrogen Energy 2019, 44, (5), 3088-3098.
  3. Lou, X.; Liu, Z.; Hou, J.; Zhou, Y.; Chen, W.; Xing, X.; Li, Y.; Liao, Q.; Zhu, X., Modification of the anodes using MoS2 nanoflowers for improving microbial fuel cells performance. Catal. Today 2021, 364, 111-117.
  4. Zou, L.; Lu, Z. S.; Huang, Y. H.; Long, Z.-e.; Qiao, Y., Nanoporous Mo2C functionalized 3D carbon architecture anode for boosting flavins mediated interfacial bioelectrocatalysis in microbial fuel cells. J. Power Sources 2017, 359, 549-555.
  5. Zeng, L.; Zhao, S.; Zhang, L.; He, M., A facile synthesis of molybdenum carbide nanoparticles-modified carbonized cotton textile as an anode material for high-performance microbial fuel cells. RSC Advances 2018, 8, (70), 40490-40497.
  6. Tahir, K.; Miran, W.; Jang, J.; Maile, N.; Shahzad, A.; Moztahida, M.; Ghani, A. A.; Kim, B.; Lee, D. S., MnCo2O4 coated carbon felt anode for enhanced microbial fuel cell performance. Chemosphere 2021, 265, 129098.
  7. Lascu, I.; Locovei, C.; Bradu, C.; Gheorghiu, C.; Tanase, A. M.; Dumitru, A., Polyaniline-Derived Nitrogen-Containing Carbon Nanostructures with Different Morphologies as Anode Modifier in Microbial Fuel Cells. Int. J. Mol. Sci. 2022, 23, (19), 11230.
  8. Pu, K.-B.; Zhang, K.; Guo, K.; Min, B.; Chen, Q.-Y.; Wang, Y.-H., Firmly coating carbon nanoparticles onto titanium as high performance anodes in microbial fuel cells. Electrochim. Acta 2021, 399, 139416.
  9. Wang, Y.; Li, B.; Cui, D.; Xiang, X.; Li, W., Nano-molybdenum carbide/carbon nanotubes composite as bifunctional anode catalyst for high-performance Escherichia coli-based microbial fuel cell. Biosens Bioelectron 2014, 51, 349-355.
  10. Liu, X.; Zhao, X.; Yu, Y.-Y.; Wang, Y.-Z.; Shi, Y.-T.; Cheng, Q.-W.; Fang, Z.; Yong, Y.-C., Facile fabrication of conductive polyaniline nanoflower modified electrode and its application for microbial energy harvesting. Electrochim. Acta 2017, 255, 41-47.

Point 3: In the preparation, Why PAN was chosen, will any other polymer work??

Response: Many thanks for the valuable suggestion! With regard to electrospun carbon fibers, the most widely used precursor material is PAN, owing much to its outstanding characteristics, such as high carbon yield, commercial viability, superior mechanical properties, and the ease of obtaining uniform carbon fibers[11-13]. PAN was chosen as the precursor of electrospinning for the aforementioned reasons. We believe that other polymers, such as PVP (Polyvinyl pyrrolidone), can also work.

  1. Zhang, B.; Kang, F.; Tarascon, J.-M.; Kim, J.-K., Recent advances in electrospun carbon nanofibers and their application in electrochemical energy storage. Progress in Materials Science 2016, 76, 319-380.
  2. Peng, S.; Li, L.; Kong Yoong Lee, J.; Tian, L.; Srinivasan, M.; Adams, S.; Ramakrishna, S., Electrospun carbon nanofibers and their hybrid composites as advanced materials for energy conversion and storage. Nano Energy 2016, 22, 361-395.
  3. Kopeć, M.; Lamson, M.; Yuan, R.; Tang, C.; Kruk, M.; Zhong, M.; Matyjaszewski, K.; Kowalewski, T., Polyacrylonitrile-derived nanostructured carbon materials. Prog. Polym. Sci. 2019, 92, 89-134.

Point 4: Do the authors studied the effect of changing the electrospinning parameters on the structure or morphology of the produced fibers and therefore their effect on the performance?

Response: Thanks very much for the excellent suggestion. The fiber structure or morphology will be affected by electrospinning parameters such as the viscosity of the spinning precursor solution, voltage, temperature, and distance, which have been reported in the literature[14-16]. In this work, we mainly discuss the performance variation of molybdenum-doped carbon nanofibers with different Mo contents as anode materials in microbial fuel cells. To control the experimental variables, we kept the electrospinning parameters constant. The effect of electrospinning parameters on fiber structure or morphology is the focus of our study in the next work.

  1. Yoon, J.; Yang, H. S.; Lee, B. S.; Yu, W. R., Recent Progress in Coaxial Electrospinning: New Parameters, Various Structures, and Wide Applications. Adv. Mater. 2018, 30, (42), e1704765.
  2. Kong, L.; Ziegler, G. R., Quantitative relationship between electrospinning parameters and starch fiber diameter. Carbohydr. Polym. 2013, 92, (2), 1416-1422.
  3. Ali, A. A.; El-Hamid, M. A., Electro-spinning optimization for precursor carbon nanofibers. Composites Part A: Applied Science and Manufacturing 2006, 37, (10), 1681-1687.

Point 5: Did the authors measure the SSA of the produced material and what about the long stability test??

Response: Many thanks for the excellent corrections! Molybdenum-doped carbon nanowires in this work are non-porous materials, according to SEM results, and our work focuses on the surface modification of carbon nanofibers by molybdenum doping, so we did not measure the SSA. We have performed a long stability test, as shown in Figure 5a.

Reviewer 2 Report

Interesting research work with quality presentation.

some minor question: 

Scheme 1 is low quality and not has to much information 

row 115 and later 2K kilo prefix is k not K

there are no information about the variance of measurement and in case precise data like row 253 (1287.38 mW m-2,) it would be nessesary to evaluate the results.

Author Response

Dear Reviewers,

I would like to express my deep thanks to you for your constructive suggestions and comments for improving the manuscript. According to your comments, we carefully revised the manuscript and highlighted the changes in blue color. The detail responses to your comments are as follows. If further revisions are needed, please let us know.

With best regards,

Dr Xiaoshuai Wu

Suzhou University of Science and Technology, China

Email: wuxiaoshuai365@163.com

Interesting research work with quality presentation.

some minor question: 

Point 1: Scheme 1 is low quality and not has to much information 

Response: Many thanks for the excellent corrections! We modified the scheme 1 in row 95. The detailed preparation steps of the material are in “2.1. Preparation of Mo-doped carbon nanofibers”.

Point 2: row 115 and later 2K kilo prefix is k not K

Response: Many thanks for the valuable suggestion! We have revised “K” to “k” in row 121 and row 234.

Point 3: there are no information about the variance of measurement and in case precise data like row 253 (1287.38 mW m-2,) it would be nessesary to evaluate the results.

Response: Many thanks for the reviewer’s comments and suggestions. We added variance information to the data at row 253 in Figure 5 b.

Reviewer 3 Report

In the work Electrospinning Mo-doped carbon nanofibers as an anode to simultaneously boost bioelectrocatalysis and extracellular electron transfer in microbial fuel cells, Xiaoshuai Wu, Xiaofen Li, Zhuanzhuan Shi, Xiaohai Wang, Zhikai Wang, Chang Ming Li, characteristics of the Mo-doped carbon nanofiber (Mo-CNFs) anodic process using flavin mononucleotide (FMN) and S. putrefaciens CN32 as an electroactive microorganism with lactic acid was described. The system was characterized using various techniques. The Mo-doped carbon nanofiber modification itself is not a significant novelty. Various Mo-modified carbon materials used in the construction of in microbial fuel cells can be found in the literature. Plenty of materials have already been used for full microbial fuel cells construction. What's new in this publication.

It is a pity that a complete microbiological system with a cathode has not been constructed. This is a pretty big shortcoming of this article.

Was the Mo-doped carbon nanofiber also tested on the cathode?

Please provide a table with other Mo doped carbon materials that have been used in the construction of the anode in microbial fuel cells. Is the tested system better than those published in the literature in terms of current density, cell power, or charge accumulation.

After some minor adjustments, I recommend publishing it in Materials

Author Response

Dear Reviewers,

I would like to express my deep thanks to you for your constructive suggestions and comments for improving the manuscript. According to your comments, we carefully revised the manuscript and highlighted the changes in blue color. The detail responses to your comments are as follows. If further revisions are needed, please let us know.

With best regards,

Dr Xiaoshuai Wu

Suzhou University of Science and Technology, China

Email: wuxiaoshuai365@163.com

Point 1: In the work Electrospinning Mo-doped carbon nanofibers as an anode to simultaneously boost bioelectrocatalysis and extracellular electron transfer in microbial fuel cells, Xiaoshuai Wu, Xiaofen Li, Zhuanzhuan Shi, Xiaohai Wang, Zhikai Wang, Chang Ming Li, characteristics of the Mo-doped carbon nanofiber (Mo-CNFs) anodic process using flavin mononucleotide (FMN) and S. putrefaciens CN32 as an electroactive microorganism with lactic acid was described. The system was characterized using various techniques. 

The Mo-doped carbon nanofiber modification itself is not a significant novelty. Various Mo-modified carbon materials used in the construction of in microbial fuel cells can be found in the literature. Plenty of materials have already been used for full microbial fuel cells construction. What's new in this publication.

Response: Many thanks to the reviewers for their valuable suggestions and comments. As far as we know, it is a novel method to prepare Mo-doped carbon nanofibers by electrospinning technology for use in the anode of microbial fuel cells. Electrospinning is a direct and simple method for producing continuous nanofibers on a large scale. Fabrication of Mo-doped carbon nanofibers by electrospinning is expected to realize the large-scale preparation of efficient anode materials and promote the commercialization of microbial fuel cells.

Point 2: It is a pity that a complete microbiological system with a cathode has not been constructed. This is a pretty big shortcoming of this article.

Was the Mo-doped carbon nanofiber also tested on the cathode?

Response: Many thanks for the reviewer’s comments and suggestions. In this work, we constructed an H-type double chamber apparatus (including anode and cathode compartments) to test the performance of Mo-doped carbon nanofibers as anodes in microbial fuel cells. The topic of this work is the anode of microbial fuel cells, so we don’t test Mo-doped carbon nanofiber on the cathode.

Point 3: Please provide a table with other Mo doped carbon materials that have been used in the construction of the anode in microbial fuel cells. Is the tested system better than those published in the literature in terms of current density, cell power, or charge accumulation.

After some minor adjustments, I recommend publishing it in Materials.

Response: Many thanks for the valuable suggestion! We add the Table S1(Supporting Information) to compare the performance of the anode materials prepared by this work with other Mo doped carbon materials and previously reported anode materials, such as follows:

Anode materials

Reactor

configuration

Substrate

Source of

inoculation

Performance

Pmax (mW m-2)

Reference

Mo-CNFs ΙΙ

Dual chamber

M9 + Lactate

S.putrefaciens CN32

1287.38 mW m-2

This work

Fe@CNFs-20

Dual chamber

M9 + Lactate

S.putrefaciens CN32

731.05 mW m-2

[1]

CNTs/CNFs

Dual chamber

artificial wastewater + PBS

mixed

bacteria

362 ± 20 mW m-2

[2]

GL-MoS2/CC

Dual chamber

PBS + sodium acetate

mixed bacteria

960.4mW m−2

[3]

Mo2C@CF

Dual chamber

Lactate

S.putrefaciens CN32

1052mW m−2

[4]

Mo2C/CCT

Single chamber

sodium acetate

mixed bacteria

1120 mW m−2

[5]

MnCo2O4@CF

Dual chamber

sodium lactate

mixed bacteria

945 mW m−2

[6]

PANI-T-900

Dual chamber

Wastewater + acetate

mixed bacteria

40.4 mW m−2

[7]

CNPs/Ti

Single chamber

sodium acetate

mixed bacteria

616 mW m−2

[8]

Mo2C/CNTs

Single chamber

PBBM + glucose

E. coli

1050±26.4 mW m−2

[9]

pristine CC

Dual chamber

M9+sodium lactate

Shewanella oneidensis MR-1

51.5 mW m−2

[10]

References:

  1. Sun, X.; Wu, X. S.; Shi, Z. Z.; Li, X.; Qian, S.; Ma, Y.; Sun, W.; Guo, C.; Li, C. M., Electrospinning iron-doped carbon fiber to simultaneously boost both mediating and direct biocatalysis for high-performance microbial fuel cell. J. Power Sources 2022, 530, 231277.
  2. Cai, T.; Huang, M. H.; Huang, Y. X.; Zheng, W., Enhanced performance of microbial fuel cells by electrospinning carbon nanofibers hybrid carbon nanotubes composite anode. Int. J. Hydrogen Energy 2019, 44, (5), 3088-3098.
  3. Lou, X.; Liu, Z.; Hou, J.; Zhou, Y.; Chen, W.; Xing, X.; Li, Y.; Liao, Q.; Zhu, X., Modification of the anodes using MoS2 nanoflowers for improving microbial fuel cells performance. Catal. Today 2021, 364, 111-117.
  4. Zou, L.; Lu, Z. S.; Huang, Y. H.; Long, Z.-e.; Qiao, Y., Nanoporous Mo2C functionalized 3D carbon architecture anode for boosting flavins mediated interfacial bioelectrocatalysis in microbial fuel cells. J. Power Sources 2017, 359, 549-555.
  5. Zeng, L.; Zhao, S.; Zhang, L.; He, M., A facile synthesis of molybdenum carbide nanoparticles-modified carbonized cotton textile as an anode material for high-performance microbial fuel cells. RSC Advances 2018, 8, (70), 40490-40497.
  6. Tahir, K.; Miran, W.; Jang, J.; Maile, N.; Shahzad, A.; Moztahida, M.; Ghani, A. A.; Kim, B.; Lee, D. S., MnCo2O4 coated carbon felt anode for enhanced microbial fuel cell performance. Chemosphere 2021, 265, 129098.
  7. Lascu, I.; Locovei, C.; Bradu, C.; Gheorghiu, C.; Tanase, A. M.; Dumitru, A., Polyaniline-Derived Nitrogen-Containing Carbon Nanostructures with Different Morphologies as Anode Modifier in Microbial Fuel Cells. Int. J. Mol. Sci. 2022, 23, (19), 11230.
  8. Pu, K.-B.; Zhang, K.; Guo, K.; Min, B.; Chen, Q.-Y.; Wang, Y.-H., Firmly coating carbon nanoparticles onto titanium as high performance anodes in microbial fuel cells. Electrochim. Acta 2021, 399, 139416.
  9. Wang, Y.; Li, B.; Cui, D.; Xiang, X.; Li, W., Nano-molybdenum carbide/carbon nanotubes composite as bifunctional anode catalyst for high-performance Escherichia coli-based microbial fuel cell. Biosens Bioelectron 2014, 51, 349-355.
  10. Liu, X.; Zhao, X.; Yu, Y.-Y.; Wang, Y.-Z.; Shi, Y.-T.; Cheng, Q.-W.; Fang, Z.; Yong, Y.-C., Facile fabrication of conductive polyaniline nanoflower modified electrode and its application for microbial energy harvesting. Electrochim. Acta 2017, 255, 41-47.

Reviewer 4 Report

The authors presented a paper in which they proposed a method to enhance the performance of a microbial fuel cell based on Shewanella putrefaciens by modifying anode with Mo2C nanoparticles. A special aspect of the work is that Mo-doped carbon nanowires were made using the electrostatic spinning technique. The study is rather interesting, but there are some minor comments

1. In "Materials and methods" authors use "M9 buffer" abbreviation. Not everyone is familiar with it's composition so authors should add that. 

2.  The formula for potassium ferricyanide is not the correct one.

3. The authors claim that they were able to observe an increase in the number of lattice defects in Mo-CNFs using Raman spectroscopy (Fig 2B). Perhaps it would be worth trying to confirm this assumption with some additional method, such as atomic force microscopy?

4. In a discussion of biofilms on the surface of the CNFs anode, the authors state that the microorganisms on the surface of the CNFs anode were scarce, but make no suggestion as to the mechanism by which the addition of molybdenum nanoparticles could improve the biocompatibility of the anode. I feel like this section should be slightly expanded.

Nevertheless, the work itself is quite well done, and I think it is worthy of being published in Materials

Author Response

Dear Reviewers,

I would like to express my deep thanks to you for your constructive suggestions and comments for improving the manuscript. According to your comments, we carefully revised the manuscript and highlighted the changes in blue color. The detail responses to your comments are as follows. If further revisions are needed, please let us know.

With best regards,

Dr Xiaoshuai Wu

Suzhou University of Science and Technology, China

Email: wuxiaoshuai365@163.com

The authors presented a paper in which they proposed a method to enhance the performance of a microbial fuel cell based on Shewanella putrefaciens by modifying anode with Mo2C nanoparticles. A special aspect of the work is that Mo-doped carbon nanowires were made using the electrostatic spinning technique. The study is rather interesting, but there are some minor comments

Point 1: In "Materials and methods" authors use "M9 buffer" abbreviation. Not everyone is familiar with it's composition so authors should add that. 

Response: Many thanks for the valuable suggestion! We add the composition of M9 buffer in row 106.

Point 2: The formula for potassium ferricyanide is not the correct one.

Response: Many thanks for the reviewer’s comments and suggestions. We modified the formula of potassium ferricyanide to K3Fe[(CN)6] in row 119.

Point 3: The authors claim that they were able to observe an increase in the number of lattice defects in Mo-CNFs using Raman spectroscopy (Fig 2B). Perhaps it would be worth trying to confirm this assumption with some additional method, such as atomic force microscopy?

Response: Many thanks for the reviewer’s comments and suggestions. We believe Mo-CNFs are amorphous based on the XRD test results. We tried to use atomic force microscopy to determine the lattice defects, but no lattice was found, so we used Raman to analyze the lattice defects of carbon materials.

Point 4: In a discussion of biofilms on the surface of the CNFs anode, the authors state that the microorganisms on the surface of the CNFs anode were scarce, but make no suggestion as to the mechanism by which the addition of molybdenum nanoparticles could improve the biocompatibility of the anode. I feel like this section should be slightly expanded.

Nevertheless, the work itself is quite well done, and I think it is worthy of being published in Materials

Response: Thanks for the reviewer's comments. We further explain the mechanism by which addition of molybdenum nanoparticles improves the biocompatibility of the anode in row 249-252.